# Genome-Wide Analysis of the Polygalacturonase Gene Family Sheds Light on the Characteristics, Evolutionary History, and Putative Function of *Akebia trifoliata*

**DOI:** 10.3390/ijms242316973

**Published:** 2023-11-30

**Authors:** Xiaoxiao Yi, Wei Chen, Ju Guan, Jun Zhu, Qiuyi Zhang, Huai Yang, Hao Yang, Shengfu Zhong, Chen Chen, Feiquan Tan, Tianheng Ren, Peigao Luo

**Affiliations:** Key Laboratory of Plant Genetics and Breeding, Sichuan Agricultural University, Chengdu 611130, China; yxx1ao@163.com (X.Y.); spongefarmer@163.com (W.C.); 18383521900@163.com (J.G.); 18383574542@163.com (J.Z.); zhangqiuyi0802@163.com (Q.Z.); yanghuai@stu.sicau.edu.cn (H.Y.); 2020101002@stu.sicau.edu.cn (H.Y.); zhongsicau@gmail.com (S.Z.); icbrcc2018@163.com (C.C.); tanfq@sicau.edu.cn (F.T.); renth@sicau.edu.cn (T.R.)

**Keywords:** *Akebia trifoliata*, fruit cracking, polygalacturonase, evolutionary relationship, expression profiling

## Abstract

Polygalacturonase (PG) is one of the largest families of hydrolytic enzymes in plants. It is involved in the breakdown of pectin in the plant cell wall and even contributes to peel cracks. Here, we characterize *PGs* and outline their expression profiles using the available reference genome and transcriptome of *Akebia trifoliata*. The average length and exon number of the 47 identified *AktPGs,* unevenly assigned on 14 chromosomes and two unassembled contigs, were 5399 bp and 7, respectively. The phylogenetic tree of 191 *PGs*, including 47, 57, 51, and 36 from *A. trifoliata*, *Durio zibethinus*, *Actinidia chinensis,* and *Vitis vinifera*, respectively, showed that *AktPGs* were distributed in all groups except group G and that 10 *AktPGs* in group E were older, while the remaining 37 *AktPGs* were younger. Evolutionarily, all *AktPGs* generally experienced whole-genome duplication (WGD)/segmental repeats and purifying selection. Additionally, the origin of conserved domain III was possibly associated with a histidine residue (H) substitute in motif 8. The results of both the phylogenetic tree and expression profiling indicated that five *AktPGs*, especially *AktPG25,* could be associated with the cracking process. Detailed information and data on the *PG* family are beneficial for further study of the postharvest biology of *A. trifoliata*.

## 1. Introduction

Fruit ripening is a complex physiological and biochemical process, usually accompanied by changes in color, texture, and flavor, and in some fleshy fruits, even by cracking [1,2]. In many plants, fruit cracking has evolved as a sophisticated dispersal mechanism for spreading seeds and extending their range [3]. However, cracking can lead to low marketability, higher storage costs, and large economic losses in some fleshy fruits [4]. Understanding the details of cracking is very important for effectively solving the problem in commercial fruit production.

Cracking is common in legumes, with cracks developing along the ventral and dorsal sutures at maturity [5]. This cracking facilitates a range of production activities, such as threshing and seeding [6]. Cracking also occurs in a few fleshy species, such as *Durio zibethinus* [7], *Prunus avium* [8], and *Citrullus lanatus* [9]. Fleshy fruit cracking can be classified into two basic categories: noninherent and inherent. Noninherent cracking, such as that found in *C. lanatus* and *Malus domestica*, is occasionally caused by environmental factors. Inherent cracking, such as in *D. zibethinus* [10] and *Akebia trifoliata* [11], is one of the typical characteristics of a given species and is a programmed physiological process controlled by genetic regulation. In addition, the fruit of *A. trifoliata* also experiences “August cracking”, which are natural and regular crack along its ventral suture during its ripening season, so it is an ideal material for investigating the molecular mechanism responsible for inherent cracking.

Various factors, such as physiological status, genetic components, and environmental changes, can lead to cracking in the peel [12], which, as a mechanical barrier, plays an irreplaceable role in protecting the whole fruit against biotic or abiotic stress [2]. However, as the fruit grows and develops, the biochemical characteristics of the peel also change synchronously. When pressure inside the tissue is higher than the mechanical resistance of the peel cell wall, cracks occur [13,14]. Since pectin is a common and major component of the primary cell wall and intercellular layer, respectively, and plays an important role in intercellular adhesion [15,16], this may indicate that pectin hydrolysis could be the essential process underlying fruit cracking. Therefore, identifying the genetic component associated with pectin disassembly is very important in order to further understand the molecular mechanism of fruit cracking.

Among hydrolases, polygalacturonase (PG) is one of the largest families and is also the most widely studied class of pectin hydrolases [17,18]. PG plays a central role in pectin degradation by catalyzing the breakdown of α-(1,4)-galacturonic bonds in the pectin molecule [19] and is classified into endo-PGs, exo-PGs, and rhamnose-PGs according to its mode of action on pectin [18]. To date, the PG gene family has been identified in several species, including *Zea mays* [20], *C. lanatus* [21], *Pyrus bretschneideri* [22], *Actinidia chinensis* [23], and *Vitis vinifera* [24]. Various studies have suggested that PG genes are widely involved in plant growth and development processes such as seed germination [25], organ abscission [26], pod cracking [27], anther dehiscence [28], and fruit softening [22].

*A. trifoliata*, commonly known as augmelon and wild banana [11], is a perennial woody vine mainly distributed in East Asian countries such as China, Japan, and Korea [29] and is becoming a new fruit crop in various regions of China due to its high nutritional and medicinal value [30,31]. Previous studies have suggested that the fruit of *A. trifoliata* physiologically belongs to a typical respiratory climacteric fruit and naturally cracks along the ventral suture line in the late stage of fruit development [11,32]. In addition, high temperatures and rain are usually accompanied by the harvesting of *A. trifoliata* fruits. They contribute to the fruit’s deterioration through contamination by biotic and abiotic factors, largely increasing the difficulty of storage and significantly shortening its shelf life. Clearly, cracking would be a major limiting factor for the commercial exploitation of *A. trifoliata* as a fresh fruit crop.

In the past, only a few studies have explored the cracking issue of *A. trifoliata* fruit. The results of a comprehensive transcriptomic and proteomic analysis suggested that the cracking of *A. trifoliata* fruit was associated with cell wall structural changes and rearrangements [33], and this view was subsequently supported by the results of a metabolic analysis [34]. At the same time, another report showed that the degradation process of the *A. trifoliata* cell wall was highly associated with PG activity [35], which indicated that PG could largely regulate fruit cracking. However, the molecular mechanism by which *PGs* regulate fruit cracking is still unknown.

During the ripening of *A. trifoliata*, *PGs* may play a significant role in fruit cracking. However, we still lack knowledge on *PGs* in *A. trifoliata*. To provide basic information to further elucidate the detailed regulatory mechanism, we planned to systemically outline the structural profile, such as the component, number, size, cis-acting element, chromosomal position and conserved domain of putative proteins of *PGs*, to investigate the evolutionary events experienced by *PGs,* and to determine their expression level using available genomic and transcriptomic data. The results would be helpful to study the cracking trait of *A. trifoliata* in the future.

## 2. Result

### 2.1. Systemic Identification of PGs in the A. trifoliata Genome

The results of both BLAST and BLASTP analyses consistently showed that there were 47 *AktPGs* in the *A. trifoliata* genome. They were named *AktPG1* to *AktPG47* according to their chromosomal locations. Physically, only *AktPG46* and *AktPG47* were assigned to unassembled contig00874 and contig00909, respectively (Table 1), while the other 45 *AktPGs* were unevenly distributed on the 16 chromosomes, except for chromosomes 6 and 15. In addition, they were mainly distributed in the end region of the chromosomes. The number of *AktPGs* on chromosomes 5, 10, and 16 was the highest, at six, while that on chromosomes 2, 7, 12, and 13 was only one (Figure 1).

In terms of genetic structure, the average length and exon number of all 47 *AktPGs* were 5398.7 bp, varying from 1650 to 28,733 bp, and 7, varying from 4 to 10, respectively. The average length, isoelectric point (PI), molecular weight (MW), and instability index of the putative proteins were 440.42, 7.54, 47.75 kDa, and 35.02, respectively (Table 1). Evolutionarily, the results of intraspecies collinearity analysis showed that twenty-three (48.94%), eleven (23.40%), ten (21.28%), and three (6.38%) *AktPGs* were produced by whole-genome duplication (WGD) or segmental duplication, dispersed, tandem, and proximal duplication, respectively.

### 2.2. Phylogenetic Tree of AktPGs

The phylogenetic tree, consisting of 47 *AktPGs* for *A. trifoliata*, 57 *DzPGs* for *D. zibethinus*, 51 *AcPGs* for *A. chinensis,* and 36 *VvPGs* for *V. vinifera,* showed that all 191 *PGs* were clustered into seven branches from A to G (Figure 2). In groups A, B, D, and E, there were similar PG numbers or similar percentages of *PGs* per species. In group C, the number of *DzPGs* was only three; moreover, there were no *VvPGs* in group F. Group G consisted of only two genes (*AcPG15* and *DzPG49*), neither being *AktPG* or *VvPG*.

### 2.3. Interspecific Collinearity

Through interspecific collinearity, we detected a total of 156 homogenous gene pairs between *A. trifoliata* and the other three species (Figure 3), which consisted of 66 pairs between 26 (55.32%) *AktPGs* and 36 (63.16%) *DzPGs*, 59 pairs between 26 (55.32%) *AktPGs* and 36 (70.59%) *AcPGs,* and 31 pairs between 24 (51.06%) *AktPGs* and 18 (50%) *VvPGs*.

### 2.4. Conserved Structures and Motifs of Putative AktPGs

The results of multiple sequence ratios and conserved structural domain analyses revealed that most AktPGs contained four conserved structural domains (I, II, III, and IV) associated with protein catalytic and binding sites (Figure 4). The conservation degree of structural domain III was obviously lower than those of structures I, II, and IV. In addition, 10 AktPGs (AktPG1, AktPG7-9, AktPG21, AktPG31, AktPG32, AktPG35, AktPG43, and AktPG45) were usually absent in the conserved structural domain III. In contrast, five AAs, including the fourth and sixth in conserved structural domain I, the first and third AAs in conserved domain II, and the third AA in conserved structure domain IV, did not show any variation in any of the 47 AktPGs.

The results of conserved motif analyses showed that the number of conserved motifs ranged from six (only AktPG25) to nine. Conserved motif 1 covered conserved structural domains I and II, and motif 3 and motif 4 covered conserved structural domains IV and III, respectively (Table 2). The numbers of AktPGs containing seven, eight, and nine motifs were three, nineteen, and twenty-four, respectively (Figure 5). However, conserved motifs 1, 2, 3, and 6 widely existed in all 47 AktPGs. The rates of motifs 4, 5, 7, 9, and 10 reached 0.79, 0.64, 0.94, 0.91, and 0.89, respectively. Motif 8 only existed in all 10 (21.3%) AktPGs of group E. Moreover, AktPGs in groups A, C, and D had identical motif compositions, while those in groups B, E, and F had different motif compositions.

### 2.5. Ka/Ks Value of Homologous AktPG Pairs

Ka/Ks value is the ratio of the nonsynonymous substitution rate (Ka) to the synonymous substitution rate (Ks) of two protein-coding genes, which is often used to determine whether there is selective pressure acting on protein-coding genes, thus reflecting the evolutionary selection of the species. A total of 467 homologous *AktPG* pairs were detected, in which only one gene pair (*AktPG15* and *AktPG17*) had a Ka/Ks value (1.122) larger than 1, while all values of the other homologous *AktPG* pairs were less than 0.5 (Appendix A), which indicated that *AktPGs* mainly experienced purifying selection.

### 2.6. Cis-Acting Elements of the AktPGs

A total of 745 cis-acting elements were identified within the 2000 bp upstream region of the *AktPG* promoter (Figure 6), and they could be classified into two types with 12 subtypes: hormone-responsive with five subtypes (abscisic acid (ABA)-, auxin-, gibberellin (GA)-, methyl jasmonate (MeJA)-, and salicylic acid (SA)-responsive elements) and environment-responsive with seven subtypes (anaerobic induction, defense and stress response, light response, physiological rhythm, drought induction, low-temperature response, and damage response). MeJA-responsive elements and light-responsive elements were the most hormone- and environment-responsive elements and accounted for 39% and 54% of the total, respectively. In addition, 40 (85%) and 45 (96%) *AktPGs* had ABA-responsive cis-acting and light-responsive cis-acting elements, respectively.

The number of cis-acting elements each *AktPG* exhibited showed a large variation, from four (*AktPG16*) to thirty-three (*AktPG17*); similarly, the number of subtypes that each *AktPG* contained also showed a large variation, from three (*AktPG16* and *AktPG12*) to eleven (*AktPG17*). In addition, although the numbers and types of cis-acting elements of *AktPGs* exhibited large variation, all *AktPGs* simultaneously contained at least one hormone-responsive and one environment-responsive subtype.

### 2.7. Expression Profiles of AktPGs in Two Independent Transcriptome Datasets

The expression level of more than 30 *AktPGs* was either undetectable or very low in the two transcriptome datasets, and there were only 16 and 15 *AktPGs* with an FPKM value detectable in one sample in the tissue- and development-related transcriptome (Figure 7a) and disease- and development-related transcriptome (Figure 7b) of *A. trifoliata*, respectively. In addition, 14 *AktPGs* were simultaneously expressed in the two transcriptome datasets, although their expression profiles were not completely the same. Both *AktPG22* and *AktPG33* exhibited a high expression level in tissue- and development-related transcriptome data, while only *AktPG42* exhibited high expression in disease- and development-related transcriptome data. Moreover, *AktPG22* and *AktPG33* exhibited tissue- and developmental-stage-specific expression, while *AktPG42* also exhibited differential expression among samples with different disease resistance levels. We further found that *AktPG22* and *AktPG33* only exhibited high expression levels in the later stages of both flesh and seed.

In fact, we also would like to note the *AktPGs* that exhibited developmental-stage-specific expression in two transcriptome datasets, namely *AktPG5*, *AktPG7*, *AktPG11*, *AktPG32*, and *AktPG44*. Among the five genes, the expression levels of *AktPG5*, *AktPG32,* and *AktPG44* generally showed a gradual decrease, while that of *AktPG7* showed a gradual increase, with developmental progress in the two transcriptome datasets. However, although the expression level of *AktPG11* also showed a gradual decrease with developmental progress in tissue- and development-related transcriptome data, the disease- and development-related expression profiles showed obvious differences among different samples with different disease resistance levels.

## 3. Discussion

Previous publications have suggested that *PGs*, mainly involved in pectin degradation by hydrolysis, are a very ancestral gene family and exist in almost all land plants, especially flowering plants [19,36]. However, *PGs* of different species possibly experienced different events in evolutionary processes that contributed to their current structural and functional characteristics. Therefore, we are mainly concerned with the possible evolutionary events, structural characteristics, and functional divergences of *PGs* in *A. trifoliata*.

### 3.1. Putative Evolutionary Events Experienced by AktPGs

At present, *PGs* have been widely identified in various species from algae to angiosperms [37], such as *Chlamydomonas reinhardtii* [36], *Z. mays* [20], and *Populus* [38]. However, reports examining basal eudicot *PGs* as an evolutionarily important branch because of their role as a bridge between basal angiosperms and core eudicots (comprising 80% extant land plants) have been very few [39]. Recently, omics data have supported, especially through the various versions of the genome and different tissue transcriptomes of *A. trifoliata*, their being typical representatives of basal eudicots [40], which provided opportunities to systemically investigate the possible evolutionary history of *AktPGs*.

In the present study, the *A. trifoliata* genome referred to is that reported by [41] in 2022 because the long terminal repeat (LTR) assembly index of this version is as high as 11.9, and it could be truly called the reference genome according to the assembly quality standard suggested by [42]. Moreover, only the corresponding files of that version were available. A total of 47 *AktPGs* were identified (Table 1), which was generally fewer than that in many core eudicots [37,38]. Both the selection style and duplication type of members are the two most important factors when we investigate the evolutionary history of a given gene family. Here, we only found one homogenous gene pair (*AktPG15* and *AktPG17*) with a Ka/Ks value (1.122) larger than 1 among a total of 467 homologous gene pairs, while all the others were less than 1 or even 0.5 (Appendix A), which suggested that *AktPGs* as well as *AktNBSs* [43], *AktMADS-boxs* [44], *AktWRKYs* [45], *AktNACs* [46], *AktSODs* [47], and *AktDofs* [48] experienced strong purifying selection in the evolutionary process.

However, *AktPGs*, similar to *AktNBSs* [43], were produced by more duplication types compared with the other reported gene families in *A. trifoliata* (Table 1, Appendix A), which indirectly supported the view that *AktPGs* could be an ancestral gene family. In addition, this interpretation also agreed with the results of both interspecific collinearity (Figure 3) and phylogenetic tree analysis (Figure 2) because genes in the E group have been confirmed to be essential and indispensable in almost all plants of different species [20,36]. Comprehensively, *AktPGs* could be of ancestral origin, and they may have experienced both various duplication types and strong purifying selection.

### 3.2. A Histidine Residue (H) Substitute Could Influence the Structural Characteristics of AktPGs

The structural domains of proteins are closely related to their functions [49]. Significant differences in amino acid sequences among members of PGs typically contain four conserved domains [50]. The core amino acid sequences of domains I and II are SPNTDG and GDDC, respectively, in which three aspartic acid residues (D) form the catalytic site [51]. The core amino acid sequence of domain III is CGPGHG, and histidine residues (H) are thought to be involved in the catalytic reaction [52]. Domain IV, consisting of RIK, interacts with the carboxylic acid groups in the substrate and binds to the substrate [53].

Here, sequence comparison found that domains I, II, and IV widely existed in each AktPG, while domain III was absent in all 10 AktPGs of the E group (Figure 4). Moreover, there was at least one completely identical AA in domains I, II, and IV among all 47 AktPGs (Figure 4). Obviously, domains I, II, and IV were highly conserved, while the conservation of domain III was obviously low. By further comparison, we found a significant difference in the fifth histidine residue (H) involved in the catalytic reaction between the 10 AktPGs in the E group and the other 37 AktPGs in the domain III sequence [52], and among all 37 AktPGs in domain III, the fifth AA had a consistent H, while the 10 AktPGs in group E outside of domain III did not have H, so H could be the hallmark of domain III.

Similarly, the other interesting result of sequence analysis was that motif 8 was also only distributed in all 10 AktPGs in the E group, while motif 4, covering domain III specifically, existed in the other 37 AktPGs. The next problem is determining which motif is older. Various studies have confirmed that PGs in group E are widely distributed in plant species and are the oldest [36,54,55], so motif 8 could be older than motif 4. In addition, the significantly lower exon number (5~7) of the 10 *AktPGs* in the E group compared with that (6~10) of most *AktPGs* also further reinforces this view because gene family evolution usually results in an increase in exon number [56]. Finally, the high expression ratio (90%) of the 10 *AktPGs* and the low ratio (18.9% and 16.2% in the tissue- and development-related transcriptome and disease- and development-related transcriptome, respectively) of the 37 *AktPGs* also provide persuasive evidence for this view because the last member of the same gene family is usually absent in the corresponding function due to a short evolutionary history [57,58].

To further identify the possible ancestor of the 37 younger *AktPGs* among the 10 older *AktPGs* in group E, we compared the Ka/Ks values between each member of the 10 *AktPGs* in group E and the other 37 *AktPGs* one by one. The results showed that among the 10 older *AktPGs*, *AktPG9* had the smallest average Ka/Ks value (0.23) among the other 37 *AktPGs*, and the Ka/Ks value ranged from 0.03 (*AktPG9* and *AktPG12*) to 0.39 (*AktPG9* and *AktPG16*). Likewise, among the 37 *AktPGs*, *AktPG12* also had the smallest average Ka/Ks value (0.21) among the 10 *AktPGs* in the E group. Considering this together, it is reasonable to assume that a histidine residue (H) substitute in motif 8 of a given member among the 10 *AktPGs* in group E further produced the 37 *AktPGs* within domain III, and that the process putatively occurred in *AktPG9* and consequently produced *AktPG12* as the first *AktPG* within domain III.

### 3.3. Several Putative Candidate AktPGs Possibly Associated with Fruit Cracking in A. trifoliata 

The results of cis-acting element analysis revealed that each *AktPG* contained at least one hormone and one abiotic stress-related response element at the same time (Figure 6), which indicated that *AktPGs* would be differentially expressed during plant development, especially in various environmental conditions. In highly expressed *AktPGs*, nine *AktPGs* out of the ten *AktPGs* in group E exhibited high expression levels (Figure 7), but we would like to exclude them as candidate genes associated with fruit cracking because genes in group E, as the oldest *PGs*, would have been responsible for common biological processes rather than species-specific physiological behavior [21]. In addition, *A. trifoliata* still retains both asexual and sexual reproduction, yet the effectiveness of sexual reproduction, mainly by seeds, is enhanced, which is generally facilitated by fruit cracking behavior [59]. Newly produced physiological behavior usually accompanies newly functionalized members rather than older members of the same gene family.

Among the remaining highly expressed *AktPGs*, only the five simultaneously highly expressed genes (*AktPG5*, *AktPG11*, *AktPG19,* and *AktPG44* in group A and *AktPG25* in group B) could be associated with fruit cracking in *A. trifoliata* because developmental change was the common factor of the two transcriptome datasets (Figure 7). Previous studies have suggested that genes in groups A and B have species-specific functions [60]. For example, *ADPG1* (*At3g57510*), *ADPG2* (*At2g41850*), *QRT2* (*At3g07970*) [27], and *DzPG25* [61], which are responsible for anther dehiscence in *Arabidopsis* and fruit cracking in *D. zibethinus*, respectively, belong to group B. Therefore, we would like to recognize *AktPG25* in group B as the candidate gene associated with fruit cracking in *A. trifoliata*, which also agreed well with the expression change in *AktPG25* (Figure 7). Confirming the functional relationship between the putative candidate genes, especially *AktPG25*, and fruit cracking in *A. trifoliata* is our next objective in the future.

## 4. Materials and Methods

### 4.1. Identification and Characterization Analysis of AktPGs

Genome sequence annotation files of *A. trifoliata* (accession IDs: GWHBISH00000000) were downloaded from the National Genomics Data Centre (https://ngdc.cncb.ac.cn/; accessed on 21 March 2023) [45]. The hidden Markov model of conserved PG sequences, glycosyl hydrolase family 28 (PF00295), was downloaded from the Pfam database (http://pfam-legacy.xfam.org/; accessed on 21 March 2023) to identify *AktPGs*, and then the sequences of all the proteins of *A. trifoliata* were scanned by HMMER 3.0 using the HMM with an E value of 1 × 10^−5^. At the same time, amino acid sequences of *Arabidopsis* PG were downloaded from the TAIR website (https://www.arabidopsis.org/; accessed on 21 March 2023) and used to identify *AktPGs*. A local BLASTP was conducted to find each putative candidate with a similarity > 50%, identity > 30%, query coverage > 95%, and E-value < 1 × 10^−10^. The physicochemical properties of putative AktPG were calculated using the Expasy Protoparam website (https://web.expasy.org/protparam/; accessed on 28 March 2023). The gene replication events among *A. trifoliata PGs* were analyzed using multiple collinear scanning toolkits (MCScanX) with the default parameters.

### 4.2. Phylogenetic Analysis, Selective Pressure, and Collinearity of AktPGs

The reference genome sequences of *D. zibethinus* (accession IDs: PRJNA407962) and *A. chinensis* (accession IDs: ASM966300v1) as well as of *V. vinifera* (genome version 2.1) were downloaded from the NCBI website (https://www.ncbi.nlm.nih.gov/; accessed on 3 April 2023) and the Ensembl Plants website (https://plants.ensembl.org/index.html; accessed on 3 April 2023), respectively. The phylogenetic tree of all *AktPGs* and 144 reference *PGs*, including 57 *DzPGs* of respiratory climacteric and cracking *D. zibethinus*, 51 *AcPGs* of respiratory climacteric and non-cracking *A. chinensis,* and 36 *VvPGs* of neither climacteric nor cracking *V. vinifera*, was constructed using MEGA 11 software (v11.0.10, Auckland, New Zealand) according to a previously reported program [62]. To display the evolutionary selection pressure between *AktPG* pairs, the Ka/Ks ratio was calculated by TBtools-II software (v2.012, Chengjie Chen, China). Similarly, we also performed a collinearity analysis using TBtools software.

### 4.3. Gene Structure and Conserved Motif Analysis

Multiple sequence comparisons of PGs were performed using DNAMAN software (version8, San Ramon, CA, USA) and the conserved AktPG motifs were identified using the MEME website (https://memesuite.org/me-me/tools/meme; accessed on 6 April 2023), in which the maximum motif parameter was set to 10, while the remaining parameters were default values. The final schematic diagram of the gene structure and conserved motifs of AktPGs was drawn using TBtools software.

### 4.4. Analysis of the Cis-Acting Elements of AktPGs

The possible cis-acting elements within the 2000 bp upstream region of the start codon of the *AktPG*s were predicted using the PlantCARE website (https://bioinformatics.psb.ugent.be/webtools/plantcare/html/; accessed on 13 April 2023), and the predicted results were then visualized using TBtools software.

### 4.5. Expression Analysis of the AktPGs

Growth and development transcriptome data (accession IDs: PRJNA671772) of *A. trifoliata* were downloaded from the NCBI website (https://www.ncbi.nlm.nih.gov/; accessed on 10 April 2023) [48]. These data included four different developmental stages (immature, enlargement, coloring, and mature stages) in three different tissues (flesh, seeds, and peels). Disease transcriptome data (accession IDs: PRJCA014987) of *A. trifoliata* were downloaded from the National Genomics Data Centre (https://ngdc.cncb.ac.cn/; accessed on 11 July 2023). Transcriptome data included three different treatment objects (control peel group, disease-resistant peel group, and disease-susceptible group) at three developmental stages (May, June, and July). FPKM values were extracted from the transcriptome data using Hisat2 software (v2.1.0, Mihaela Pertea, USA) and DESeq2 (v1.36.0, Michael I Love, Germany) to estimate gene expression levels. TBtools software was used to construct a heatmap of *AktPG* expression.

## 5. Conclusions

A total of 47 *AktPGs* were identified in the *A. trifoliata* reference genome, of which 45 (95.7%) were unevenly assigned to 14 high-quality assembled pseudochromosomes. Evolutionarily, they were mainly produced by WGD/segmental repeats and purifying selection. Further phylogenetic analysis classified them into six groups (A–F), and 10 *AktPGs* belonging to group E were the oldest group. We found a mutually exclusive relationship between motif 4 and motif 8 among 47 *AktPGs*, and after carefully comparing the sequences, we would like to suggest that a histidine residue substitute in motif 8 could be highly associated with the origin of conserved structural domain III of *AktPGs* in motif 4. Together, we speculated that five *AktPGs* (*AktPG5*, *AktPG11*, *AktPG19*, *AktPG44,* and *AktPG25*), especially *AktPG25*, could be associated with the cracking process. The results, including data, gene resources, and conclusions, will be helpful to further genetically improve the traits of interest in *A. trifoliata* in the future.

## Figures and Tables

**Figure 1 ijms-24-16973-f001:**
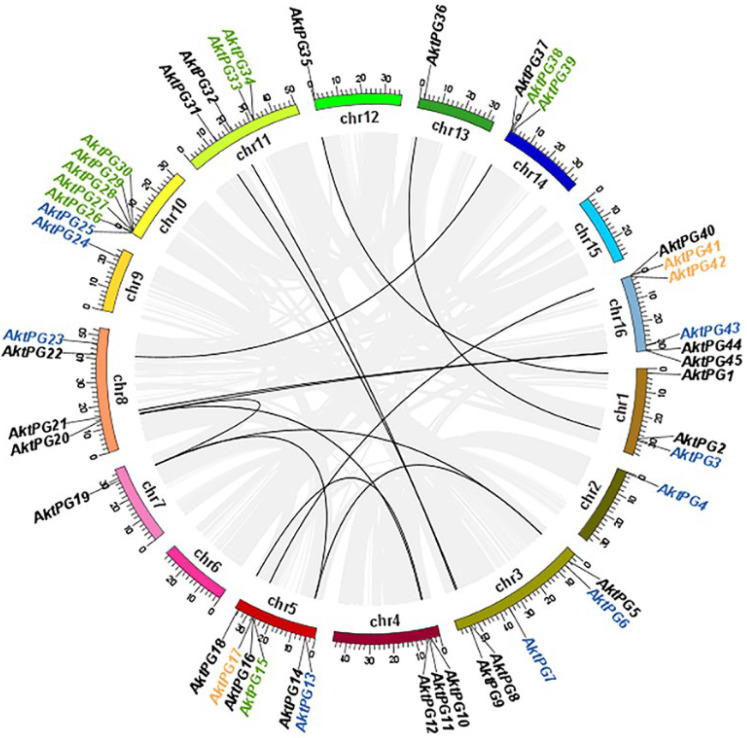
Chromosomal distribution of *PGs* in *Akebia trifoliata*. Different segments represent different chromosomes. Gray and black lines represent segmental duplication pairs in the whole *A. trifoliata* and between *AktPG* pairs, respectively. Genes marked in black, blue, green, and yellow were putatively produced by whole-genome duplication (WGD) or segmental, dispersed, tandem, and proximal duplication, respectively.

**Figure 2 ijms-24-16973-f002:**
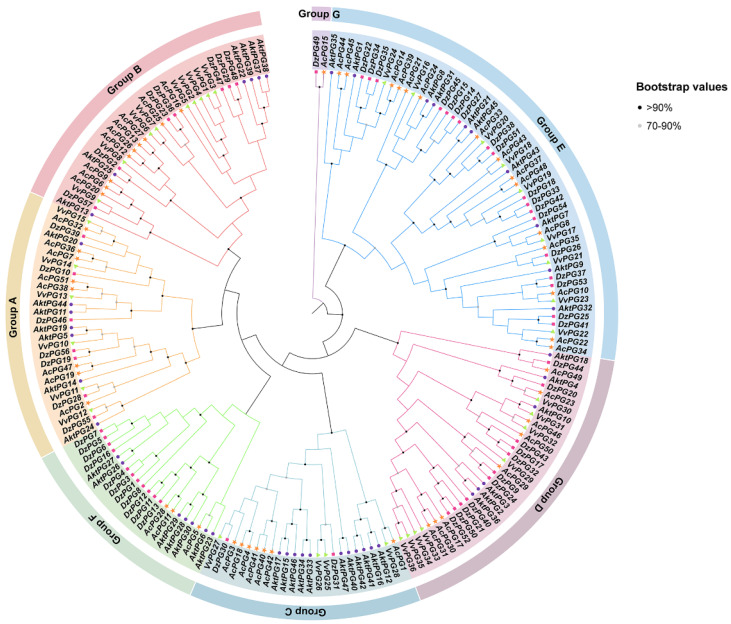
Phylogenetic tree of *PGs* in *A. trifoliata*, *Durio zibethinus*, *Actinidia chinensis*, and *Vitis vinifera*. Different shapes represent different species. According to the tree, all *AktPGs* were divided into six groups.

**Figure 3 ijms-24-16973-f003:**
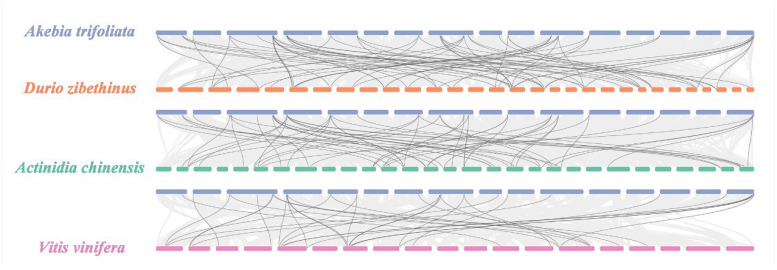
*PGs* collinearity analysis among *A. trifoliata* and *D. zibethinus*, *A. chinensis,* and *V. vinifera*. Black lines represent collinearity among different *PGs*.

**Figure 4 ijms-24-16973-f004:**
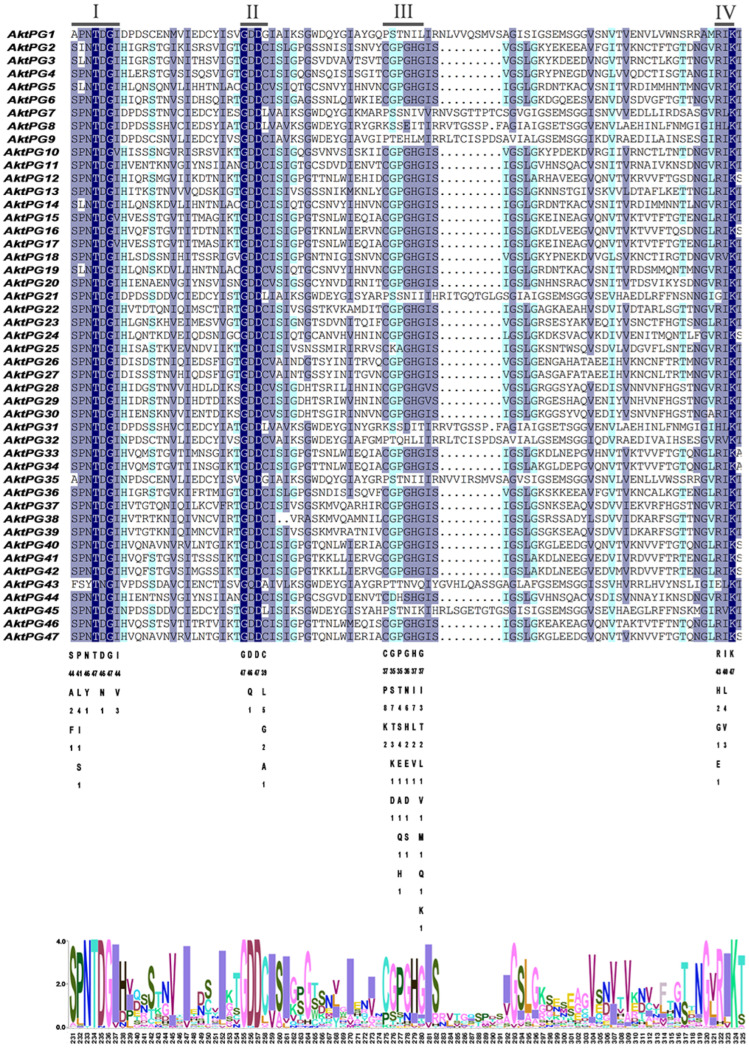
Multiple sequence alignment of AktPGs. Underlining indicates the conserved domains of AktPG*s*: I (SPNTDGI), II (GDDC), III (CGPGHG), and IV (RIK). The shade of the color represents the degree of sequence similarity.

**Figure 5 ijms-24-16973-f005:**
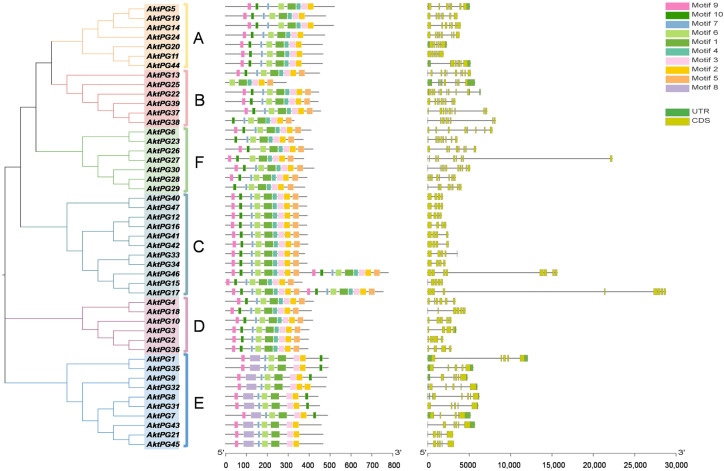
Gene structures and conserved motifs of *AktPGs*. The left side indicates the phylogenetic tree of 47 AktPGs. The middle shows the distribution of conserved motifs of 47 AktPGs. The right side shows the exon/intron structures of *AktPGs*. UTR, CDS, and black lines represent noncoding regions, coding regions, and introns, respectively.

**Figure 6 ijms-24-16973-f006:**
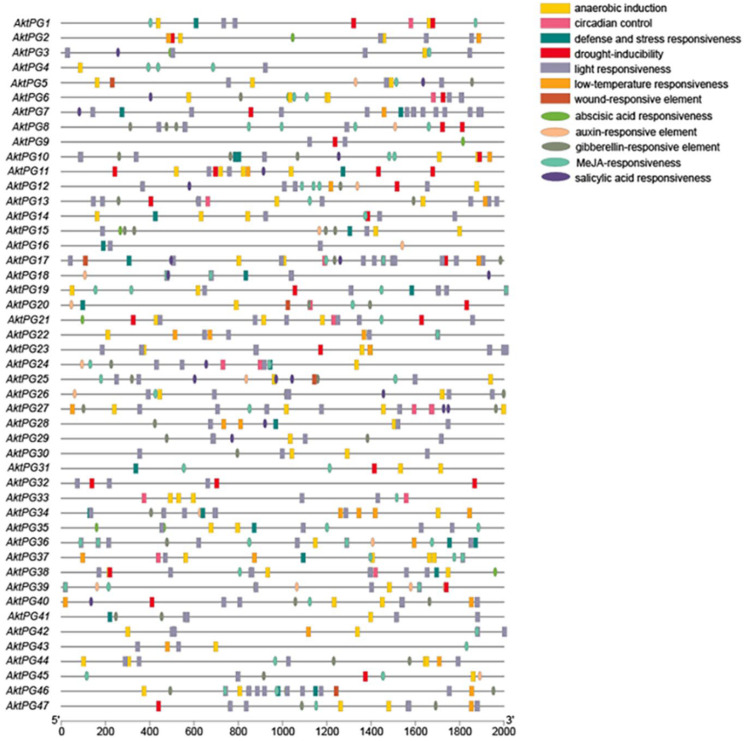
Distribution of cis-acting elements within the 2000 bp upstream region of *AktPGs*. Different icons represent different types of promoter cis-acting elements.

**Figure 7 ijms-24-16973-f007:**
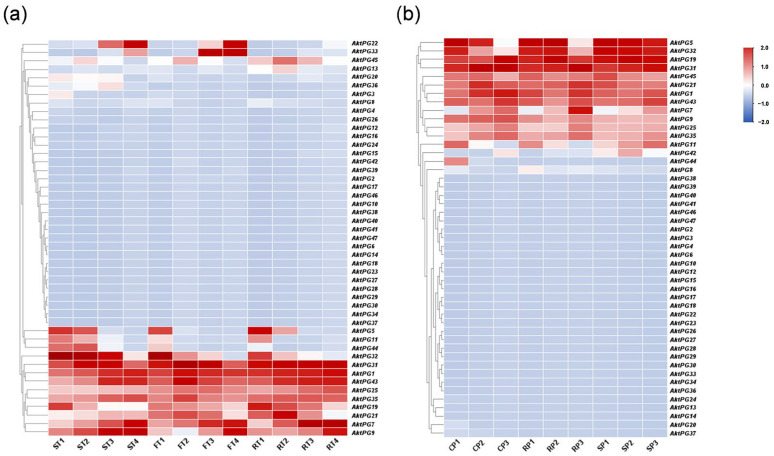
Expression profiles of *AktPGs* in different stages. The degree of change from blue to red indicates the intensity of the expression level. (**a**) Expression profiles of *AktPGs* in tissue- and development-related transcriptomes. ST, FT, RT and 1, 2, 3, 4 represent three different tissues (seed, flesh, and peel tissues) and four different developmental stages (immature, enlargement, coloring, and mature stages) of *A. trifoliata*, respectively. (**b**) Expression profiles of *AktPGs* in disease- and development-related transcriptomes. CP, RP, SP and 1, 2, 3 represent three different treatment objects (control peel group, disease-resistant peel group, and disease resistant–susceptible mixed pool) and developmental stages (May, June, and July) of *A. trifoliata*, respectively.

**Table 1 ijms-24-16973-t001:** Detailed characteristics of both *AktPGs* and putative AktPGs.

Gene Name	Chromosome	Location		Gene Length (bp)	Exons	Duplication Type	Putative Protein			
							Length (AA)	PI	Molecular Weight (kDa)	Instability Index
*AktPG1*	1	1950986	1963052	12,066	5	WGD or Segmental	491	8.62	55.26	45.59
*AktPG2*	1	29549713	29551549	1836	4	WGD or Segmental	396	8.76	42.22	29.29
*AktPG3*	1	30682795	30686212	3417	4	Dispersed	399	4.94	42.26	30.34
*AktPG4*	2	579257	582555	3298	6	Dispersed	420	8.80	44.25	27.41
*AktPG5*	3	3762763	3767823	5060	7	WGD or Segmental	520	7.54	56.88	43.59
*AktPG6*	3	5777237	5785021	7784	8	Dispersed	408	7.56	44.56	36.70
*AktPG7*	3	35682662	35687812	5150	5	Dispersed	487	7.13	53.12	36.22
*AktPG8*	3	53310300	53316508	6208	7	WGD or Segmental	442	9.23	48.59	39.23
*AktPG9*	3	54139078	54143861	4783	6	WGD or Segmental	484	4.75	53.24	43.41
*AktPG10*	4	3864384	3867211	2827	5	WGD or Segmental	416	9.24	44.77	36.83
*AktPG11*	4	4589231	4591100	1869	6	WGD or Segmental	466	4.67	50.71	41.14
*AktPG12*	4	5079077	5080727	1650	4	WGD or Segmental	390	9.27	42.28	32.39
*AktPG13*	5	3716964	3722125	5161	10	Dispersed	449	5.33	48.69	31.54
*AktPG14*	5	3742665	3746619	3954	7	WGD or Segmental	516	6.85	56.43	47.09
*AktPG15*	5	27256023	27257798	1775	5	Tandem	366	5.35	38.32	23.56
*AktPG16*	5	27294394	27296597	2203	4	WGD or Segmental	388	8.80	41.64	31.14
*AktPG17*	5	27307604	27336337	28,733	8	Proximal	753	5.71	79.99	23.59
*AktPG18*	5	34397399	34401928	4529	6	WGD or Segmental	410	8.16	43.94	35.70
*AktPG19*	7	28823177	28826730	3553	7	WGD or Segmental	478	7.17	51.78	44.81
*AktPG20*	8	13692797	13695055	2258	6	WGD or Segmental	463	6.29	50.55	42.41
*AktPG21*	8	15213064	15216086	3022	6	WGD or Segmental	466	4.97	51.03	36.10
*AktPG22*	8	41653829	41660180	6351	10	WGD or Segmental	446	6.08	48.93	38.24
*AktPG23*	8	46547108	46550654	3546	9	Dispersed	371	7.52	40.24	32.55
*AktPG24*	9	23425160	23428978	3818	8	Dispersed	474	9.11	51.99	44.54
*AktPG25*	10	529791	535457	5666	9	Dispersed	290	8.24	31.81	43.07
*AktPG26*	10	592482	598293	5811	8	Tandem	417	6.34	45.67	35.91
*AktPG27*	10	606091	628366	22,275	9	Tandem	374	7.88	39.75	27.74
*AktPG28*	10	644584	647904	3320	8	Tandem	389	7.06	42.12	36.69
*AktPG29*	10	651049	655078	4029	8	Tandem	378	8.71	41.58	46.00
*AktPG30*	10	655783	660884	5101	10	Tandem	423	8.14	45.57	36.50
*AktPG31*	11	13500656	13506726	6070	5	WGD or Segmental	449	6.55	49.18	40.01
*AktPG32*	11	21482203	21488172	5969	6	WGD or Segmental	479	6.07	52.11	34.72
*AktPG33*	11	31436526	31440085	3559	5	Tandem	378	7.96	39.89	18.05
*AktPG34*	11	31477315	31479413	2098	4	Tandem	390	9.05	41.41	19.24
*AktPG35*	12	325546	331005	5459	5	WGD or Segmental	490	9.45	55.23	45.09
*AktPG36*	13	1986765	1989582	2817	4	WGD or Segmental	394	9.10	42.04	26.42
*AktPG37*	14	463984	471132	7148	10	WGD or Segmental	455	9.37	50.01	33.82
*AktPG38*	14	473659	481805	8146	9	Tandem	327	9.36	35.84	35.60
*AktPG39*	14	485377	488688	3311	9	Tandem	443	8.82	48.61	31.00
*AktPG40*	16	287716	289512	1796	4	WGD or Segmental	388	9.35	41.86	34.11
*AktPG41*	16	389335	391791	2456	4	Proximal	391	9.52	42.34	25.09
*AktPG42*	16	405390	407880	2490	4	Proximal	394	9.67	42.89	29.64
*AktPG43*	16	30413390	30419071	5681	6	Dispersed	457	5.14	48.75	38.24
*AktPG44*	16	32206491	32211639	5148	7	WGD or Segmental	463	4.98	49.95	46.13
*AktPG45*	16	32296843	32299964	3121	6	WGD or Segmental	466	5.34	51.38	35.44
*AktPG46*	Contig00874	8841	24460	15,619	8	Dispersed	778	7.15	82.83	19.74
*AktPG47*	Contig00909	12723	14521	1798	4	Dispersed	388	9.35	41.89	34.11

**Table 2 ijms-24-16973-t002:** Information on the conserved motifs of 47 AktPGs identified.

Motif	Wide	Best Possible Match	PG Domain
Motif1	41	ITAPGDSPNTDGIHYQSSTNVVIEDSVIKTGDDCISIGSGT	I, II
Motif2	29	GYVRNITFENITMNNVQNPIIIDQNYCPH	
Motif3	29	EAGVSNVTVKNVVFTGTTNGVRIKTWQGG	IV
Motif4	21	VWIENINCGPGHGISIGSLGK	III
Motif5	29	ISNVTYKNIKGTSATEVAIKFDCSKSVPC	
Motif6	29	PTGPTSLRFYNSKNIVISGLTSINSPQFH	
Motif7	11	GTIDGQGSVWW	
Motif8	50	LTGSFNLTSHMTLFLDKGAVILGSQDEKHWPLIDPLPSYGRGRELPGGRY	
Motif9	16	DFGAVGDGKTDDTKAF	
Motif10	15	TYLVKPTIFSGPCKS	

## Data Availability

All data are available in this article and the Appendix A.

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
