# Peer review of "Genome-Wide Analysis of the Polygalacturonase Gene Family Sheds Light on the Characteristics, Evolutionary History, and Putative Function of Akebia trifoliata"

_ijms, 2023, doi:10.3390/ijms242316973_

Round 1
Reviewer 1 Report
Comments and Suggestions for Authors
Everything in the article can be accepted, but the chapter two must be placed as number two as mentioned

Author Response
Everything in the article can be accepted, but the chapter two must be placed as number two as mentioned.
Answer: Thank you very much for your positive evaluation on our manuscript. We had revised the last paragraph of the introduction section to an introduction instead of a conclusion based on your suggestion.
Reviewer 2 Report
Comments and Suggestions for Authors
Yi et al.'s manuscript thoroughly explores the Polygalacturonase (PG) enzyme family in Akebia trifoliata (AktPG), focusing on structural and evolutionary aspects. The study investigates AktPG distribution, average length, exon number, and evolutionary relationships through a phylogenetic tree involving PGs from diverse plant species. Evolutionary mechanisms, including whole-genome duplication (WGD)/segmental repeats and purifying selection, are emphasized.
The study highlights a noteworthy observation regarding a potential association between conserved domain III origin and a histidine residue (H) substitution in motif 8, suggesting a link between sequence variations and functional implications. Notably, specific AktPGs, especially AktPG25, are implicated in the peel cracking process, supported by detailed expression profiling and phylogenetic analysis. This research advances our understanding of the AktPG family's molecular evolution and its role in A. trifoliata's postharvest biology.
In conclusion, the manuscript's comprehensive characterization of AktPGs, integrating genomic and transcriptomic data with evolutionary insights, lays a foundation for further research on A. trifoliata's postharvest biology. The findings may extend to similar enzymes in other plant species, contributing to the broader field of plant molecular biology with potential implications for agricultural and horticultural practices. The manuscript can be accepted for the publication. I have following concern about the current version of the manuscript.
#Results 2.1: The brief details about BLAST analysis should be provided here. What was the similarity cut-offs, query coverage, identity percentage cutoff and P-values. What similarity cutoff was used to select the these 47 AktPGs.
#Results 2.1, paragraph 2. The word structurally adds confusion here with protein structures. Alternative word should be used.
#Results 2.4: The described structural domains of AktPG shown be shown the consensus 3D protein structure of AktPG.
#Results 2.5: Please define Ka/Ks value
Author Response
Results 2.1: The brief details about BLAST analysis should be provided here. What was the similarity cut-offs, query coverage, identity percentage cut off and P-values. What similarity cutoff was used to select the these 47 AktPGs.
Answer: Thank you very much for your positive evaluation on our manuscript. We had supplemented the detailed methods of BLASTP in Materials and Methods 4.1 based on your suggestions, including similarity cut off, query coverage, identity percentage cut off, and E-values for candidate gene screening.
Results 2.1, paragraph 2. The word structurally adds confusion here with protein structures. Alternative word should be used.
Answer: Your suggestions had been adopted. As you commented, the word here may cause ambiguity. Therefore, we replaced it with In the genetic structure to eliminate readers' confusion.
Results 2.4: The described structural domains of AktPG shown be shown the consensus 3D protein structure of AktPG.
Answer: Thank you for your friendly comments. In fact, our research may focus more on comparing the sequence changes of conserved domains. Therefore, we want to preserve the analytical form of this protein structure. In addition, we provided a more detailed definition of the division of the four structural domains in result 2.4.
Results 2.5: Please define Ka/Ks value.
Answer: Sorry for ignoring the definition of Ka/Ks value. Based on your suggestion, we added a description of the Ka/Ks value at result 2.5.
Reviewer 3 Report
Comments and Suggestions for Authors
Comments to the Author,
The manuscript on “Genome-Wide Analysis of the Polygalacturonase Gene Family Sheds Light on the Characteristics, Evolutionary History and Putative Function in Akebia trifoliata”. Interesting research compiled on the Polygalacturonase Gene Family. Using computational tools helped in identifying polygalacturonase genes in Akebia trifoliata and also understanding phylogenetic relationships. Manuscripts can be accepted for publication.
1. Authors could have experimented with some of the 47 polygalacturonases in vitro assays. As you know the gene sequence of polygalacturonases, cDNA can be cloned into expression vectors and conduct an enzyme assay. This will enable us to know all these 47 polygalacturonases are actively involved in Akebia trifoliata.
Author Response
Authors could have experimented with some of the 47 polygalacturonases in vitro assays. As you know the gene sequence of polygalacturonases, cDNA can be cloned into expression vectors and conduct an enzyme assay. This will enable us to know all these 47 polygalacturonases are actively involved in Akebia trifoliata.
Answer: Thank you for your friendly comments on our manuscript. Your suggestion is exactly our next objective in the future.